# Effects of brand and brand trust on initial trust in fully automated driving system

**Zixin Cui**[ID][1]*, **Nianzhi Tu**[1], **Makoto Itoh**[2]

**1** Department of Risk Engineering, Graduate School of System and Information Engineering, University of Tsukuba, Tsukuba, Ibaraki, Japan, **2** Institute of Systems and Information Engineering, University of Tsukuba, Tsukuba, Ibaraki, Japan

* cui@css.risk.tsukuba.ac.jp

**Data Availability Statement:** Some data in this study cannot be shared publicly because data contain potentially identifying or sensitive brand and consumer information. Data are available from

## Abstract

Before Automated Driving Systems (ADS) with full driving automation (SAE Level 5) are placed into practical use, the issue of calibrating drivers' initial trust in Level 5 ADS to an appropriate degree to avoid inappropriate disuse or improper use should be resolved. This study aimed to identify the factors that affected drivers' initial trust in Level 5 ADS. We conducted two online surveys. Of these, one explored the effects of automobile brands and drivers' trust in automobile brands on drivers' initial trust in Level 5 ADS using a Structural Equation Model (SEM). The other identified drivers' cognitive structures regarding automobile brands using the Free Word Association Test (FWAT) and summarized the characteristics that resulted in higher initial trust among drivers in Level 5 ADS. The results showed that drivers' trust in automobile brands positively impacted their initial trust in Level 5 ADS, which showed invariance across gender and age. In addition, the degree of drivers' initial trust in Level 5 ADS was significantly different across different automobile brands. Furthermore, for automobile brands with higher trust in automobile brands and Level 5 ADS, drivers' cognitive structures were richer and varied, which included particular characteristics. These findings suggest the necessity of considering the influence of automobile brands on calibrating drivers' initial trust in driving automation.

## Introduction

Following the taxonomy and definition of driving automation by the Society of Automotive Engineers [1], an Automated Driving System (ADS) with full driving automation (Level 5) can operate vehicles under all road conditions, just as a skilled human driver, and does not require any supervision. Although Level 5 ADS is a long way from reality, its practical application is always expected since it can further optimize the traffic environment and help decrease the risk rate of traffic accidents.

To realize the practical application of Level 5 ADS in society, both technical issues and social and human issues, such as drivers' trust calibration, should be solved. Based on Lee and See's definition of human-automation trust [2], drivers' trust in driving automation is the attitude of the driver that driving automation will help achieve their driving targets in uncertain

the Laboratory for Cognitive Systems Science, Faculty of Engineering, Information, and Systems at the University of Tsukuba (contact via question@css.risk.tsukuba.ac.jp) for researchers who meet the criteria for access to confidential data.

**Funding:** This study was supported by the Japan Society for the Promotion of Science (https://www.jsps.go.jp/) KAKENHI [grant number 17KT0153] awarded to M.I. The funders had no role in study design, data collection and analysis, decision to publish, or preparation of the manuscript.

**Competing interests:** The authors have declared that no competing interests exist.

and vulnerable driving situations. When the drivers' trust in driving automation is comparable to the trustworthiness of driving automation itself, it can be considered as an appropriate calibration of trust [3]. This implies that the drivers' trust matches the actual ability of driving automation. In contrast, distrust or over-trust appear when the drivers' trust is lower or higher than the trustworthiness of driving automation, respectively [2,4]. Appropriate trust in driving automation is essential because it has a positive relationship with usage intention and social acceptance of driving automation [5]. Instead, distrust can lead to disuse [6], and therefore, can become a great obstacle to the social realization of driving automation [7,8]. However, over-trust can also lead to misuse [6], and cause fatal accidents. Level 5 ADS can work under all road conditions, just as a skilled human driver, and can reduce risk in unmanageable conditions, such as flooded roads and glare ice [1]. However, drivers' over-trust and misuse of Level 5 ADS are problems that are impossible to predict. Therefore, the appropriate calibration of drivers' trust in Level 5 ADS should be considered to ensure their appropriate use in the future.

## Development of human trust in automation

Trust is dynamic because it develops as the relationship between the trustor and trustee advances [2,9]. According to the dynamic trust framework [10], trust in Level 5 ADS develops successively via dispositional, initial, ongoing, and post-task trusts. Dispositional trust reflects a driver's propensity to trust [11,12], which depends on their culture, age, gender, personality traits [12], and early trust-related experiences [13]. Since dispositional trust develops before the first interaction with driving automation or even before the recognition of driving automation, it is not influenced by the characteristics of driving automation itself and driving contexts [12]. Initial trust refers to trust that develops from dispositional trust when the driver has just known about driving automation but has not used it. Therefore, initial trust is influenced by both dispositional trust and drivers' indirect experiences related to driving automation [10], such as information acquired from others [2]. Unlike dispositional and initial trust, ongoing and post-task trust develop by relying on direct contact with driving automation. Ongoing trust is formed during interactions based on the drivers' initial trust and perceived real-time characteristics of driving automation. Post-task trust is formed after completing an interaction with driving automation [10,11].

We aimed to focus on initial trust as it played an important role in the connection between dispositional, ongoing, and post-task trust [10,11]. Furthermore, it could affect users' initial decision of engaging in automation [2]. In Merritt and Ilgen's study on operators' trust in the Automatic Weapons Detector (AWD) [11], operators' initial trust, formed based on their first impression of the AWD, had a positive relationship with operators' post-task trust, which was measured immediately after the use of AWD. Hence, the higher the level of initial trust, the higher was the level of post-task trust, regardless of whether the level of trust changed with the first experience of automation. Therefore, besides increasing the experience of automation to adjust ongoing and post-task trust from the prior levels of trust to a more appropriate level [14,15], adjusting the level of initial trust before the first experience may be an effective way for adjusting ongoing and post-task trust, with lesser difficulty and cost. In addition, Merritt and Ilgen [11] found that initial trust was positively associated with operators' usage decisions regarding automation. Therefore, excessive and inadequate initial trust before experiencing automation may lead to initial misuse and disuse, respectively. Finally, vehicles adopting Level 5 ADS have not yet entered the market. Although the public has knowledge of Level 5 ADS, they still do not have any experience or even an adequate understanding of it. Therefore, the development of the public trust in Level 5 ADS is at the stage of *initial trust*.

## Factors that influence the formation of the trust

According to Lee and See's conceptual model of the dynamic process of trust in automation [2], prior information regarding automation can form the basis for initial trust before interaction. Analytic information helps trustors develop trust in a cognitively demanding process with rational assessments of whether automation is trustworthy. For example, reliability information [16] and analytic information provide trustors with direct evidence of the trustworthiness automation. Conversely, analogical information helps trustors develop trust in a less cognitively demanding process of assessing category membership. Analogical information such as the observed interface features of automation, acquired gossip related to automation, and perceived brand reputation, indirectly governs the formation of trust. Many empirical studies verified the influence of analytic information on trust formation [14,17]. Compared with low-reliability information, high-reliability information led to a higher level of trust in Level 5 ADS [16]. However, empirical evidence on analogical information is lacking. Although Forster et al. [16] focused on brand reputation, they found no effect on trust.

A brand is a name, symbol, trademark, or design used as an unique identifier of an individual, organization, or company in owned commodities and services to differentiate from those of its competitors [18,19]. Just as a person has a personality, a brand has a brand personality (a set of human characteristics associated with a brand), including sincerity (down-to-earth, honest, wholesome, and cheerful), excitement (daring, spirited, imaginative, and up-to-date), competence (reliable, intelligent, and successful), sophistication (upper-class and charming), and ruggedness (outdoorsy and tough) [20]. According to Celmer et al.'s framework for trust in branded autonomous vehicles [21], drivers' perceived brand personality will influence their trust and performance expectations. In other words, before judging the actual trustworthiness of an ADS, drivers may make a prior judgment regarding the identity and brand personality of the automobile brand that developed it. Therefore, analogous information that can influence the formation of initial trust in Level 5 ADS should not be limited to brand reputation. Instead, brands should be considered a whole that can convey information regarding each aspect of brand personality. However, empirical evidence is required to verify the effect of automobile brands or a perceived brand personality on drivers' trust in Level 5 ADS, which was an aim of this study.

Besides the automobile brand itself, drivers' trust in the automobile brand that developed the driving automation may be another possible influencing factor in the formation of the initial trust. Previous studies on consumer goods verified that consumers' purchase intention largely depended on their trust in the brand [22–24]. For consumer goods in an online shopping context, website brands influenced consumers' trust and perceived risk towards online retailers, and thus, in turn, influenced consumers' purchase intention [25]. Likewise, equipped with driving automation, automobiles are originally a kind of consumer good, referred to as products purchased by consumers for their enjoyment and improvement of living standards, including houses, food, and clothing [26]. Thus, when consumers plan to purchase a new car equipped with ADS, they would consider whether it was developed by a trustworthy automobile brand. However, there is limited direct evidence of the relationship between trust in automobile brands and ADS products. Therefore, the existence and extent of the impact of trust in automobile brands on trust in ADS should be examined.

## Cognition of automobile brand

If drivers' trust in ADS depends on and can be influenced automobile brands that develop it, it is necessary to discuss the kinds of automobile brands that can earn more trust in themselves and an ADS. Previous studies found that consumers' trust in a brand depended on various

factors, such as consumer satisfaction [27], perceived product quality of the brand [27,28], business behavior and climate of the employees and organization [27,29–31], and corporate credibility [32]. In addition, according to the salient value similarity (SVS) model proposed by Earle and Cvetkovich [33], people trusted and entrusted a person or an organization that possessed the same or similar viewpoints, feelings, and cognition of values. Therefore, many factors that influence brand trust were related to people's prior knowledge and information regarding their cognition of the brand. In a study on interpersonal trust among managers and professionals in organizations, McAllister [34] addressed the nature of interpersonal trust, the extent of affect-based and cognition-based trust. The level of cognition-based trust was higher than that of affect-based trust and was necessary for its development to an extent. Here, cognition-based trust refers to making a choice on whom to trust based on "good reasons" constituting evidence of trustworthiness according to prior knowledge and information [35]. Similarly, human-computer trust also includes both cognitive and affective components, although affective components are stronger indicators [36]. Accordingly, people's cognition of a trustee plays an important role in their trust in a person, organization, or machine. Furthermore, owing to different experiences, people can have different perceptions of the same brand. To better understand the relationships among drivers' trust in Level 5 ADS and automobile brands, clarifying their cognition bases of the automobile brands that develop Level 5 ADS is necessary.

**Purpose and hypotheses.** Automobile brands and drivers' trust in automobile brands have the potential to influence drivers' initial trust in Level 5 ADS. However, few studies have provided direct evidence to verify whether or to what degree these influences existed.

Therefore, this study aimed to examine whether drivers' initial trust in Level 5 ADS depended on automobile brands that developed the Level 5 ADS and can be positively influenced by drivers' trust in the automobile brands. In addition, we aimed to explore drivers' cognition characteristics of automobile brands that could earn much more initial trust in Level 5 ADS by clarifying drivers' cognitive structures on the automobile brands. To achieve these, we conducted two online surveys and our hypotheses are as follows:

Hypothesis 1 (H1): There would be a positive relationship between drivers' trust in automobile brands and their initial trust in Level 5 ADS.

Hypothesis 2 (H2): The levels of drivers' initial trust in Level 5 ADS would differ across different automobile brands.

Hypothesis 3 (H3): The cognitive structures of automobile brand drivers with higher trust and lower trust would have distinctly different characteristics.

## Survey 1

### Methods

**Procedure and participants.** An online survey in Japanese was conducted by commissioning an Internet Research Company in June 2020. In total, 206 anonymous Japanese drivers (103 men and 103 women), aged 18–69 years ($M$ = 44.6 years, $SD$ = 11.8), with an ordinary license participated (from June 8–9, 2020).

Participants were asked to complete nine questionnaires regarding nine randomly chosen automobile brands that developed driving automation. The questionnaires were presented to the participants in a random order to counterbalance the order of the automobile brands. In addition, due to ethical requests from the ethical review board of University of Tsukuba, the actual names of the brands were used for data collection; however, they were hidden under the code names like Brand 1 in this paper. Brands 1, 2, and 3 were famous automobile brands in Japan. Brand 4 was a famous automobile brand in the USA. Brands 5, 6, 7, and 8 were famous

automobile brands from Germany. Brand 9 was a famous automobile brand in Sweden. The questionnaire consisted of two sections for each automobile brand. The first and second sections used a "brand trust scale" and "human-automation trust scale" to measure drivers' trust in automobile brands and initial trust in the Level 5 ADS developed, respectively. Participants were instructed to imagine that the automobile brand would sell cars with Level 5 ADS. Before the participants completed the questionnaires, necessary explanations for Level 5 ADS were provided following descriptions from the SAE [37] (A Level 5 ADS is capable of all dynamic driving tasks including continuous lateral and longitudinal vehicle motion control and object and event detection and response. In addition, when continuous operation is difficult, Level 5 ADS can do sustained and unconditional responses without any expectation that a user will respond to a request to intervene.)

This survey was approved by the ethical review board of the Faculty of Engineering, Information and Systems at the University of Tsukuba (approval number 2020R369). Although informed consent could not be obtained since all participants were anonymously and randomly recruited in the sample pool of the Internet Research Company, all the participants were voluntary to finish the questionnaires without any prejudice and were paid according to the regulations of the Internet Research Company.

**Measurements.** The human-automation trust scale was proposed by Muir and Moray [9], which included four items: predictability, dependability, faith, and overall trust (Table 1). The brand trust scale was an eight-item measure of the first factor, "trust," of the "Japanese consumer-brand relationship scale" developed by Hatai [38] (Table 1). Both the scales were rated on a 7-point Likert scale (for brand trust, 1: I do not think so, 7: I think so; for human-automation trust: 1: Not at all, 7: Extremely high).

## Data analysis

To verify H1, structural equation modeling (SEM) was conducted using Amos version 26.0. Before the structural model was analyzed to examine the impact of trust in automobile brands on trust in Level 5 ADS, it was necessary to examine the goodness-of-fit (GOF) of the measurement model. Therefore, a confirmatory factor analysis (CFA) on the two variables was performed. Subsequently, the two measurement models were modified with modification indices [39] until a moderate GOF was obtained. After the reliability and validity of the two modified

**Table 1. Items in the trust questionnaires [9,38].**

| Scale | Item |
|---|---|
| **Trust in brand** | **TB1 I trust this brand.** |
| | **TB2 I can be in sympathy with the products and services of this brand.** |
| | **TB3 If the products come from this brand, I will be relieved.** |
| | **TB4 If someone enquired regarding this brand, I will recommend it.** |
| | **TB5 I have a good image for this brand.** |
| | **TB6 There is no failure of this brand.** |
| | **TB7 This brand will not let me down.** |
| | **TB8 The quality of this brand's products and services is excellent.** |
| **Trust in Level 5 ADS** | **TA1 To what extent can the behavior of the Level 5 ADS be predicted from moment to moment? (Predictability)** |
| | **TA2 To what extent can you count on the Level 5 ADS to do its job? (Dependability)** |
| | **TA3 To what extent will the Level 5 ADS be able to cope with other system states in the future? (Faith)** |
| | **TA4 To what extent do you trust the Level 5 ADS? (Overall trust)** |

measurement models was examined, a structural model analysis was conducted, during which the overall GOF was evaluated using the recommended model fit indices. Commonly used fit indices included the Chi-square/df, goodness-of-fit index (GFI), adjusted goodness-of-fit index (AGFI), comparative fit index (CFI), and root mean square error of approximation (RMSEA). Chi-square/df and RMSEA should be less than 5 (3 if strictly) and 0.5, respectively, and the GFI, AGFI, and CFI should be greater than 0.9 [40].

Since people of different ages and gender can develop initial trust in driving automation based on different factors, such as cognitive or emotional factors towards driving automation [41], it was also necessary to verify whether the effect of trust in automobile brands on trust in Level 5 ADS was invariant across gender and age. For sex, a multigroup invariance test was performed, and the invariance was indicated by a significant difference in the chi-square value ($p < .05$) [40] or ΔCFI, ΔGFI, ΔAGFI, and ΔRMSEA, which were not higher than 0.01 [42]. For age, the invariance was indicated by the non-significant moderating effect of age on the relationship. Therefore, a "moderation by interaction terms" method [43] was used, in which age was first multiplied with the scores of each item of trust in the automobile brand, and the subsequent combined effect was examined against trust in Level ADS.

Finally, to verify H2, a repeated-measures analysis of variance (ANOVA) was employed via SPSS version 26.0 to analyze the main effects of automobile brands on both trust in automobile brands and ADS. All data analyses were conducted using the "General Linear Model" → "Repeated measures" module. If sphericity was violated in Mauchly's test (p < .05), the degrees of freedom were corrected by Greenhouse-Geisser correction, if ε ≤ 0.75, or HuynhFeldt correction, if ε > 0.75.

## Results

**CFA and evaluation of reliability and validity.** The model fit indices of both the original measurement models did not show a moderate fit to data, trust in automobile brands: $\chi^2_{(20)} = 388.80$, *GFI* = .949, *AGFI* = .908, *CFI* = .979, *RMSEA* = .100; trust in Level 5 ADS: $\chi^2_{(2)} = 74.60$, *GFI* = .979, *AGFI* = .896, *CFI* = .990, *RMSEA* = .140. Therefore, for trust in automobile brands, items TB6, TB8, TB5, and TB2 were removed during modification to achieve a moderate GOF, $\chi^2_{(2)} = 3.19$, *GFI* = .999, *AGFI* = .996, *CFI* = 1.00, *RMSEA* = .018. For trust in the Level 5 ADS, item TA1 (predictability) was removed, and thus a moderate GOF was obtained, $\chi^2_{(0)} = .000$, *GFI* = 1.000.

The modified models showed moderate internal consistency for each factor since the composite reliability (CR) values were greater than 0.6 (Table 2). Moreover, all the factor loads of the items were greater than 0.5 and significant ($p < .01$). Both Average of Variance Extracted (AVE) and Square Multiple Correlations (SMC) values were greater than 0.5 (Table 2). These

**Table 2. Reliability and convergence validity.**

| | | Parameter significance estimation | | | | Convergence validity | | | |
|---|---|---|---|---|---|---|---|---|---|
| | | United. | S.E. | T-value | P | Std. Estimate | SMC | C.R. | AVE |
| **Trust in automobile brand** | TB1 | 1 | | | | .886 | .785 | .938 | .791 |
| | TB3 | 1.093 | 0.017 | 63.81 | *** | .944 | .891 | | |
| | TB4 | 0.947 | 0.02 | 47.388 | *** | .819 | .671 | | |
| | TB7 | 1.014 | 0.017 | 58.29 | *** | .904 | .817 | | |
| **Trust in Level 5 ADS** | TA2 | 1 | | | | .916 | .839 | .943 | .846 |
| | TA3 | 1.01 | 0.016 | 64.094 | *** | .913 | .834 | | |
| | TA4 | 1.045 | 0.016 | 66.831 | *** | .930 | .865 | | |

**Table 3. Discriminant validity.**

|  | AVE | Trust in Level 5 ADS | Trust in automobile brand |
|---|---|---|---|
| **Trust in Level 5 ADS** | .791 | **.889** |  |
| **Trust in automobile brand** | .846 | .652 | **.920** |

*Note*: The numbers in bold are the square roots of the AVE on the left. 0.652 is the correlation coefficient between trust in automobile brand and Level 5 ADS.

met the evaluation criteria of Hair et al. [40], and thus reflected the good convergence validity of the modified measurement models of trust in automobile brands and Level 5 ADS. Good discriminant validity was also considered since the square root of AVE for each factor was larger than the correlation coefficient between trust in automobile brand and Level 5 ADS (Table 3) [44].

**SEM of drivers' trust.**  Fig 1 shows the final SEM of drivers' trust with a standardized path coefficient, as well as the reported fit index values. In the measurement model, faith, dependability, and overall trust were all significant factors that positively influenced trust in the Level 5 ADS (factor load = 0.91, 0.92, and 0.93, respectively, $ps < .001$). In the structural model, trust in automobile brands positively influenced trust in Level 5 ADS (Std. estimate = 0.652, $p < .001$). For the model fit, the values of chi-square/df was approximately 5, GFI, AGFI, and CFI were greater than 0.9, and RMSEA was less than 0.05. These results indicated an acceptable model fit [40]. In addition, both the models were invariant across gender (Table 4), which showed a positive effect of trust in automobile brands on trust in Level 5 ADS for both men (Std. estimate = 0.699, $p < .001$) and women (Std. estimate = 0.599, $p < .001$). The moderating effect of age on the relationship between trust in automobile brands and Level 5 ADS was not significant (men: Std. estimate = -0.021, $p = .682$; women: Std. estimate = 0.025, $p = .573$).

**Differences in trust among automobile brands.**  The average scores of trust in automobile brands and Level 5 ADS for the nine automobile brands are shown in Fig 2.

Results of the repeated-measures ANOVA (after Greenhouse-Geisser correction) showed that the main effect of automobile brand was significant on both trust in automobile brands ($F_{(5.75, 1179.37)} = 53.17$, $p < .001$, partial $\eta^2 = 0.206$) and Level 5 ADS ($F_{(6.21, 1272.96)} = 22.74$, $p < .001$, partial $\eta^2 = 0.100$).

Post-hoc pairwise comparisons corrected by the Bonferroni method showed that for both trust in automobile brand and Level 5 ADS, the average scores for Brands 1 and 2 had no significant difference ($p = .075$; $p = .538$). However, they were significantly higher than the average scores for the other seven (Brands 3–9) ($ps < .05$). Furthermore, the average scores of both

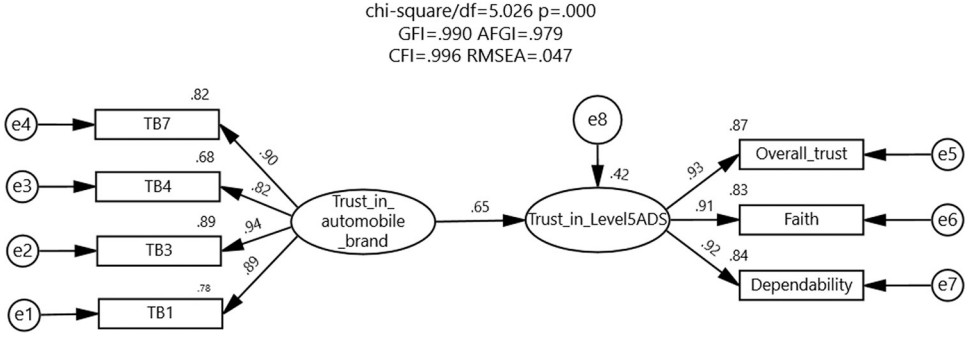

**Fig 1. SEM of drivers' trust.**

**Table 4. Measurement and structural invariance across sex.**

|  | Unconstrained | Measurement weights | Structural covariances | Measurement residuals |
|---|---|---|---|---|
| chi-square | 121.518 | 128.492 | 137.776 | 216.618 |
| df | 26 | 31 | 33 | 41 |
| *p* (GOF) | < 0.001 | < 0.001 | < 0.001 | < 0.001 |
| chi-square/df | 4.674 | 4.145 | 4.175 | 5.283 |
| GFI | 0.982 | 0.981 | 0.98 | 0.97 |
| AGFI | 0.962 | 0.966 | 0.966 | 0.959 |
| CFI | 0.993 | 0.992 | 0.992 | 0.986 |
| RMSEA | 0.045 | 0.041 | 0.041 | 0.048 |
| Δchi-square | / | 6.974 | 16.258 | 95.1 |
| Δdf | / | 5 | 7 | 15 |
| *p* (chi-square difference) | / | 0.223 | 0.023 | <0.001 |
| ΔGFI | / | -0.001 | -0.001 | -0.010 |
| ΔAGFI | / | 0.004 | <0.001 | -0.007 |
| ΔCFI | / | -0.001 | <0.001 | -0.006 |
| ΔRMSEA | / | -0.004 | -0.004 | 0.003 |

trust in automobile brand and Level 5 ADS for Brand 4 were significantly lower than those for almost all the others (Brands 1–3, 5–9) (*ps* < .01), except from the comparable result with Brand 5 (*p* = .225) for trust in Level 5 ADS. Among the average scores of both trust in automobile brand and Level 5 ADS for the rest (Brand 3, 5–9), there was almost no significant

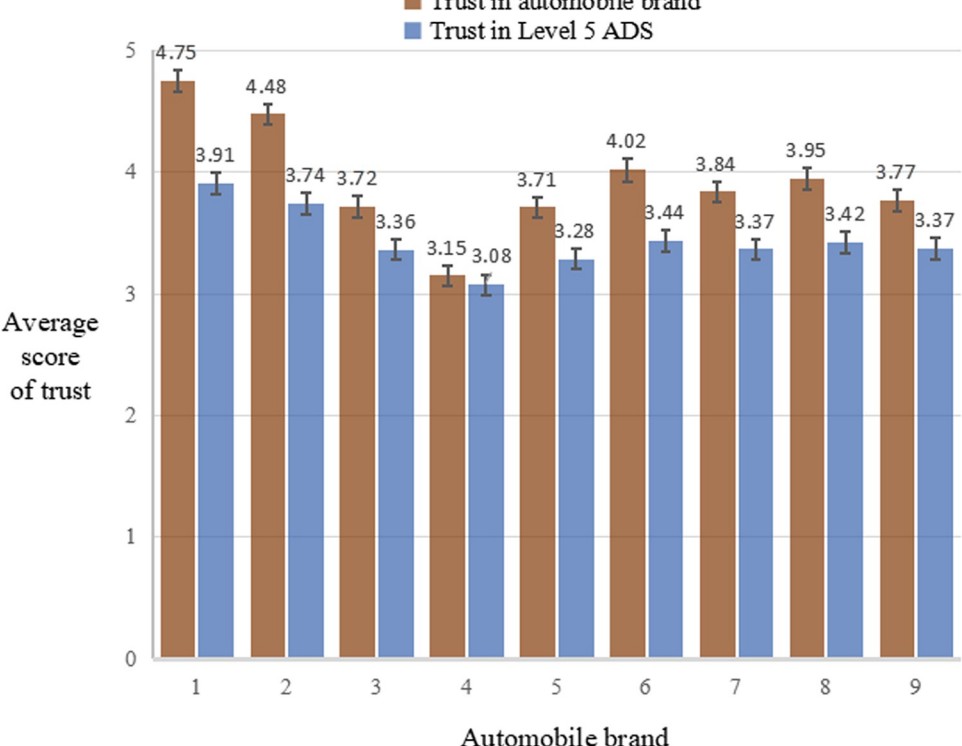

**Fig 2. Average scores of trust in automobile brand and Level 5 ADS for the nine automobile brands.** Error bars are standard errors of means.

difference ($ps > .05$), except from significant differences between Brands 5 and 6 ($p = .000$), 5 and 8 ($p = .015$), and 6 and 9 ($p = .018$) for trust in automobile brand.

## Survey 2

Results of the repeated-measures ANOVA on the main effect of automobile brand on trust showed that both trust in automobile brands and Level 5 ADS differed significantly across automobile brands. Subsequently, based on the results of the pairwise comparisons, we divided Brands 1 and 2 into a high trust group, Brand 4 into a low trust group, and the remaining six (Brands 3, 5–9) into a medium trust group. However, why some brands were highly trusted than others was still unknown. Consequently, Survey 2 was conducted to obtain cues from drivers' cognitive structures of automobile brands.

### Methods

**Participants.** An online survey in Japanese was conducted by commissioning an Internet research company in January 2021. Anonymous responses were received (from January 8–9, 2021) from 103 Japanese drivers (58 men and 45 women), aged 25–70 years (M = 50.1 years, $SD$ = 12.5), with a normal license. Same to Survey 1, all participants were anonymous and voluntary to finish the questionnaires without any prejudice, and were paid according to the regulations of the Internet Research Company. This survey was also approved by the ethical review board of the University of Tsukuba (approval number 2020R369-1).

**Data collection.** We chose Brands 1 and 2 in the high trust group, 3 and 5 in the medium trust group, and 4 in the low trust group from Survey 1 as the research objects. We used the Free Word Association Test (FWAT) to collect data, a widely used method to reveal and analyze the cognitive structure of somebody or something [45–49]. The type and number of concepts retrieved from long-term memory and the links between them could be revealed [50].

The names of the above five brands were presented individually to participants as stimulus words. According to the previous application of FWAT, participants wrote down the next word associated with the previous words that they had written down rather than the stimulus word itself [45–48,50], called the chain reaction. To avoid this risk, each stimulus word (brand name) was written five times on one page, as shown in Fig 3. Participants were asked to write down the first five words associated with a stimulus brand for an unlimited duration when each stimulant brand appeared. Similar to Survey 1, the actual name of each brand was used to collect research data; however, they were hidden under code names like Brand 1 in this paper.

**Data analysis.** To verify H3, we analyzed FWAT data based on the content analysis method, widely used in previous studies and showed high reliability [45–48,50]. Combined with the study purpose, there were six analytical processes. 1) The valid word number for each brand was counted, which referred to the total number of words written down by participants after invalid words, such as "none" or "unknown" were excluded. 2) The effective rates of the valid words was calculated using the formula [valid word number / planned word number (5) * participant number (103)]. 3) The different word number for each brand was calculated. Different word numbers represented the number of words that were different from the others among valid words. Table 5 illustrates the differences between the valid word numbers and different word numbers with an example. 4) The correlation between the level of trust and valid word numbers as well as different word numbers was analyzed. 5) The words with the same meaning or property were categorized after those that were stated only once and not associated with other words were excluded. This process was assessed once more, and some words were merged with other similar words or shifted to other categories. 6) The word number of each category was calculated and drivers' cognitive structures for each brand was drawn.

**Fig 3. Example of FWAT data collecting.**

To ensure reliability of the word categorization, two external researchers examined whether each word belonged to the current category. The consistency between the results for each word provided by the two researchers was marked as "Consensus" or "Divergence." Furthermore, the reliability was calculated by the formula [Consensus / (Consensus + Divergence) X 100] [51]. Since no word was marked "consensus" that the word did not belong to the current category, no word was moved to other categories. Finally, the intercourse reliability was 92%, which was considered reliable [52].

**Table 5. An example for the different word number and valid word number.**

| Category | Different word | Valid word number |
|---|---|---|
| **Brand personality** | 1. First-class | 3 |
| | 2. Number one in Japan | 3 |
| | 3. Number one in the world | 5 |
| **The sum** | 3 | 11 |

*Note*: This is an example of part of the data. For the category "Brand personality," three different words were obtained (the medium column); thus, the different word numbers were 3. These were stated repeatedly three, three, and five times, respectively; thus, the valid word numbers for the three different words were 3, 3, and 5, respectively, and the total valid word number was 11.

**Table 6. Valid word number, different word number, and trust.**

| Trust | Brand | Valid word numbers | Effective rate | Different word number | Trust in brand | Trust in Level 5 ADS |
|---|---|---|---|---|---|---|
| High | 1 | 163 | 31.65% | 106 | 4.75 | 3.91 |
| | 2 | 122 | 23.69% | 77 | 4.48 | 3.74 |
| Medium | 3 | 132 | 25.63% | 76 | 3.72 | 3.36 |
| | 5 | 75 | 14.56% | 46 | 3.70 | 3.28 |
| Low | 4 | 49 | 9.51% | 33 | 3.15 | 3.08 |
| Sum | | 540 | 20.97% | 338 | / | / |

## Results

**Valid word number, different word number, and trust.** Since some participants were unable to come up with all five words related to automobile brands, 540 (effective rate 20.97%) valid words were obtained for all automobile brands, with 338 different words (Table 6). Both valid and different word numbers for the high trust group ($M = 142.5$, $M = 20.5$) were higher than those for the medium trust ($M = 103.5$, $M = 13.5$) and low trust groups ($M = 48$, $M = 6$).

A correlation analysis was conducted to confirm the relationship between valid word numbers, different word numbers, and trust in automobile brands as well as trust in Level 5 ADS. Table 7 shows a significant positive correlation between valid word numbers and different word numbers and different word numbers and both trust in automobile brands as well as trust in Level 5 ADS. Furthermore, it also shows positive non-significant correlations between valid word number and both trust in automobile brand and trust in Level 5 ADS. However, Brand 3 is an exception. Although valid word numbers and different word numbers were between the two brands in the high trust group, Brand 3 belonged to the medium trust group, in which both the levels of trust in automobile brands and Level 5 ADS were lower than brands in the high trust group.

**Cognitive structure by categories.** For each automobile brand, words listed only once were deleted, and the remaining (338 words) were categorized according to their meaning or property. Fig 4 shows the categories and word numbers of each category for each automobile brand.

The remaining words were divided into seven categories: "brand personality," "product," "person," "country/region," "brand/logo," "advertising" and "others." The category of "brand personality," contained words that described participants' perceived brand personality, such as "reliable" and "safe." "Product" contained words in the name of specific products. "Person" contained names of persons related to the brand, such as the CEO or advertising spokesperson.

**Table 7. The correlation coefficient of the word number and trust.**

| | Count of category | Valid word number | Different word number | Trust in automobile brand |
|---|---|---|---|---|
| Count of category | | | | |
| Valid word number | .647 ($sig = .238$) | | | |
| Different word number | .732 ($sig = .159$) | .989** | | |
| Trust in automobile brand | .829 ($sig = .082$) | .857 ($sig = .064$) | .899* | |
| Trust in Level 5 ADS | .825 ($sig = .082$) | .874 ($sig = .053$) | .920* | .995** |

Note

** The correlation is significant at 0.01 level

* The correlation is significant at 0.05 level.

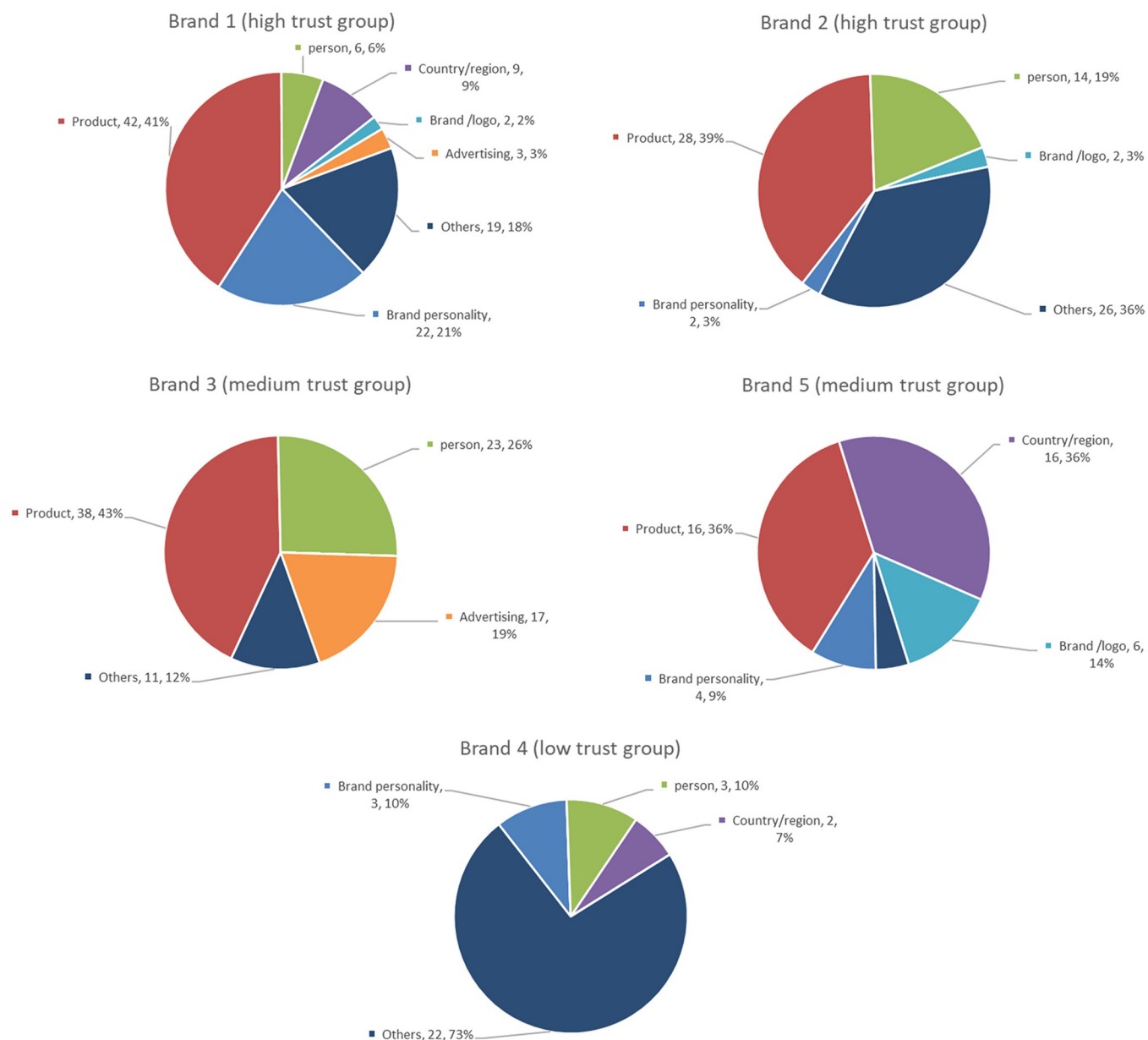

**Fig 4. Categories and the word number of each category for each automobile brand.**

For "country/region," names of countries or cities in which the brands were located were mentioned. "Brand/logo" was related to the word that the brand or logo itself was associated with, such as aliases or what the logo looked like. Finally, words from the advertising slogans were contained in the category "advertising," and words that do not belong to any categories were contained "others." To clearly understand the drivers' cognitive structure for each brand, we drew word clouds with the remaining words (Figs 5–9). However, for ethical reasons, we hid some words, such as the name of the product, that may easily lead readers to know what these brands are with mosaic. In Figs 5–9, black words on the periphery are the names of the categories, and words in different colors belong to different categories. For example, red words are the names of specific products and green words are the names of a person. For these words in

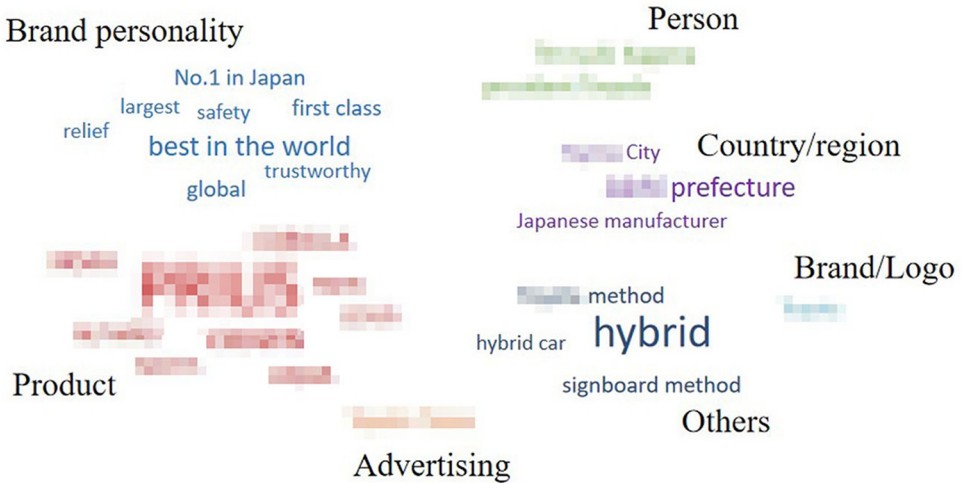

**Fig 5. Drivers' cognitive structure of Brand 1 in high trust group.**

the word clouds, the more frequently the word was associated with, the larger the font size of the word.

Importantly, the count of the category was higher for brands in the high trust group ($M = 6$) than for brands in the medium ($M = 4.5$) and low trust groups ($M = 4$). Furthermore, it was highly positively correlated with both trust in automobile brands and trust in Level 5 ADS, although it was non-significant for both (Table 7). Moreover, there was some universality and individuality in categorizing associated words among the brands. Approximately 40% of words relevant to "product" were associated with brands in the medium and high trust groups, whereas for the brand in the low trust group (Brand 4), there was no associated word relevant to "product". For all brands in the medium and high trust groups, "others" was only a small part (the largest proportion was 36% for Brand 2) or did not exist. However, the proportion of "others" reached up to 73% for Brand 4 in low trust group. Furthermore, for Brand 3 in the medium trust group, there was no "brand personality," while it existed for the other four

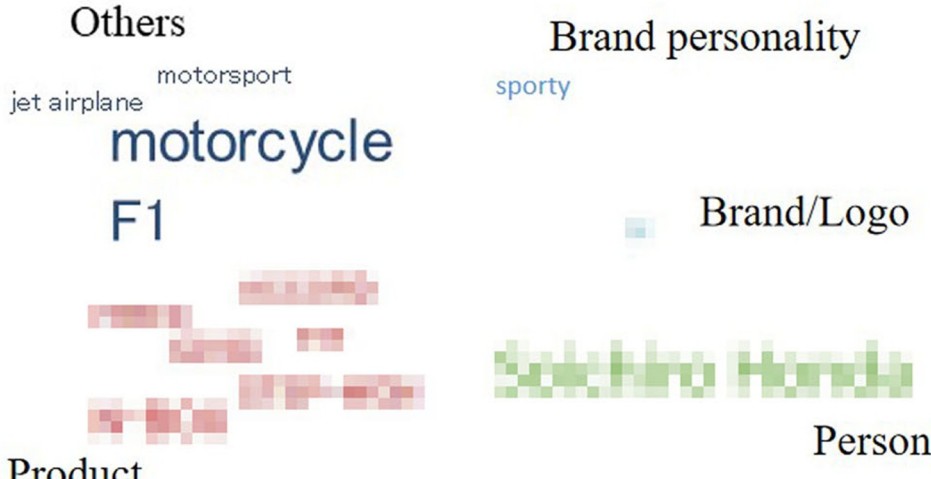

**Fig 6. Drivers' cognitive structure of Brand 2 in high trust group.**

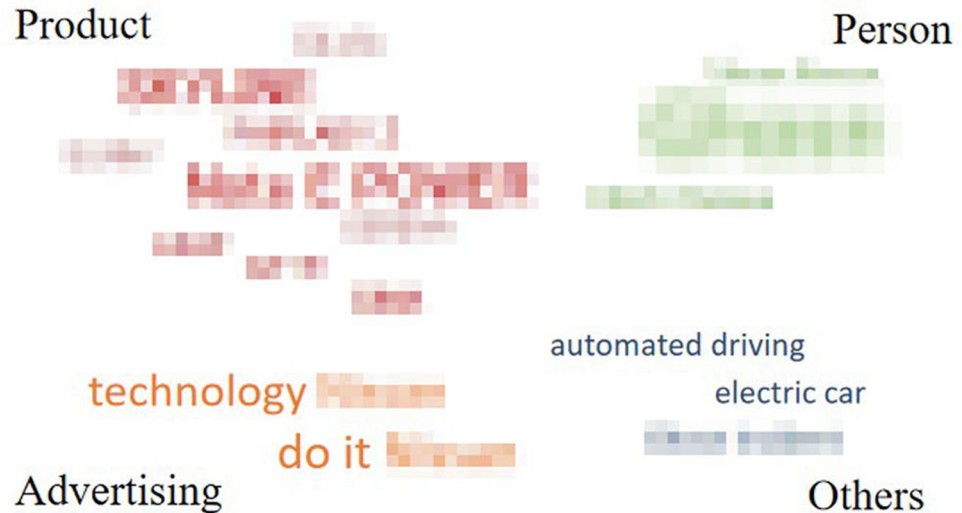

**Fig 7. Drivers' cognitive structure of Brand 3 in medium trust group.**

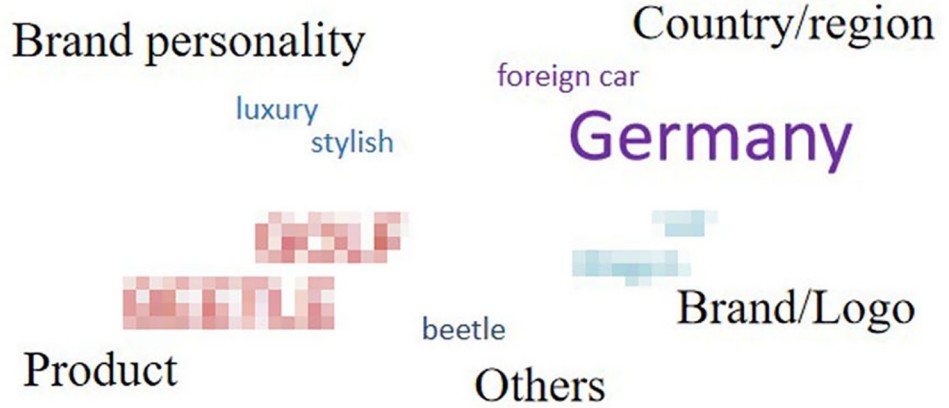

**Fig 8. Drivers' cognitive structure of Brand 5 in medium trust group.**

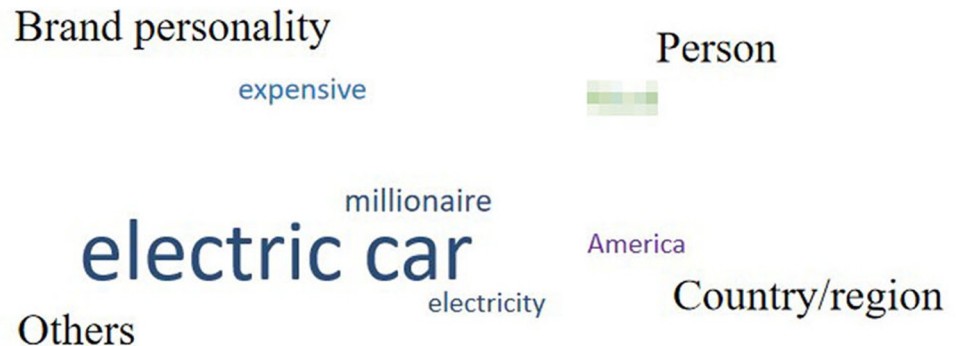

**Fig 9. Drivers' cognitive structure of Brand 4 in low trust group.**

brands. Lastly, for Brand 5 in the medium trust group, there was no "person," while it existed for the others.

**Words in cognitive structures related to unethical events.** After the numbers of associated words and drivers' cognitive structures were analyzed, the content of each associated word was examined individually so that any special cognition in drivers' cognitive structures would not be missed. Finally, for Brand 3 in the medium trust group, some associated words were relevant to unethical events that occurred recently (23.6% of the total associated words). However, there were no negative words associated with the other four brands.

## Discussion

This study explored the relationship among automobile brands, drivers' trust in automobile brands, and drivers' initial trust in Level 5 ADS. Furthermore, we clarified the common characteristics of drivers' cognitive structures regarding automobile brands that could earn higher trust in automobile brands and higher initial trust in Level 5 ADS. We hypothesized that drivers' initial trust in Level 5 ADS was different across automobile brands and positively impacted by drivers' trust in automobile brands. Moreover, drivers' cognitive structures regarding automobile brands in the high trust group had some common characteristics. These findings were discussed and integrated with previous studies to provide implications for the practical use of Level 5 ADS.

### The impact of brand trust on initial trust in Level 5 ADS

Consistent with H1, we found that when drivers did not have any direct interaction experience with Level 5 ADS, their initial trust was directly affected by the level of their trust in the automobile brand that developed it. Specifically, according to the results of Survey 1, trust in the automobile brand had a positive impact on trust in Level 5 ADS, which was invariant across gender and age. Accordingly, for automobile brands, promoting drivers' trust in automobile brands may also be a possible approach to promote drivers' initial trust in Level 5 ADS. In addition, the positive impact of trust in automobile brands on trust in Level 5 ADS implied that over-trust in automobile brands could lead to over-trust in Level 5 ADS. Meanwhile, distrust in automobile brands could lead to distrust in Level 5 ADS. Previous studies revealed the effectiveness of elaborate practice and adequate knowledge of driving automation on the calibration of drivers' initial trust in driving automation [15,53]. However, before calibrating drivers' initial trust in driving automation, their prior level of trust in the automobile brand should also be considered as a precondition.

### Effect of automobile brand on initial trust in Level 5 ADS

Since significant differences in the level of consumers' brand trust existed among brands, we examined the discrimination of the chosen nine automobile brands at the level of drivers' brand trust before we verified H2 to avoid a type I error. The results showed that drivers' trust in automobile brands was indeed different across automobile brands, which indicated the good discrimination of the chosen automobile brands. According to the subsequent results, drivers' initial trust in Level 5 ADS was also different across different automobile brands. Thus, H2 was verified. This implied that automobile brands influenced drivers' initial trust in the Level 5 ADS, which provided empirical evidence for Lee and See's conceptual model of the dynamic process of trust in automation [2] and Celmer et al.'s framework of trust in branded driving automation [21]. To adjust drivers' initial trust in the Level 5 ADS, their prior identification and judgment of the automobile brand that developed the Level 5 ADS should be

considered. Different trust calibration strategies and approaches are necessary for driving automation developed by different automobile brands.

## Drivers' cognitive structure of automobile brand

Based on the associated words analyses from automobile brands via FWAT, H3 was confirmed. Furthermore, several common characteristics of drivers' cognitive structures of automobile brands earned higher trust in automobile brands and higher initial trust in Level 5 ADS.

**Characteristic 1: Rich and varied.**   For automobile brands in the high trust group, both the number and category of associated words were far more than those in the medium and low trust groups. Therefore, for automobile brands that could earn higher trust in automobile brands and higher initial trust in Level 5 ADS, drivers had richer and more varied cognitive structures. This implied that to acquire higher trust in both the automobile brand itself and the Level 5 ADS, automobile brands should be well known and familiar to drivers, which was related to the accumulation of long-term brand promotion. Generally, familiarity was regarded as a precondition to trust somebody or something [54], and has been widely verified in different fields [55,56]. Ha and Perks [57] found that a high level of brand familiarity among consumers was a dimension of brand trust achievement.

**Characteristic 2: Have approximately 40% of the cognition of the product.**   No word relevant to "product" was associated with Brand 4 in the low trust group. In contrast, for the brands in the high and medium trust groups, many automobile products were associated. Even though the word number of "product" was quite different for each brand, the proportions of "product" in the total word number was approximately 40%. Among these associated products, many were classic automobile products with decades of history and generations of innovation, and some were special products with special functions or techniques. This indicated that the quality assurance of products played an important role in brand development. Product quality was reported to influence the level of brand trust by creating consumer satisfaction [27] and influence customers' loyalty to the brand [28].

**Characteristic 3: Have some cognition more or less related to brand personality or related person.**   No words relevant to "person" or "brand personality" were associated with brands in the medium trust group. Thus, it could be considered that for brands with medium trust, although the part of "product" existed in drivers' cognitive structures to almost the same degree as brands with high trust, there was always a missing piece, such as "person" or "brand personality." Accordingly, use of the public influence of brand spokespersons to help promote brand influence and trust seemed effective. Although attractiveness of an advertisement spokesperson seemed relatively unimportant when consumers assessed a new high-technology product [58], a better fit between spokesperson and product attributes could help better shape consumers' attitudes towards the brand [59]. In contrast, for automobile brands that could earn higher trust in the brand and higher initial trust in the products, drivers might have more cognition of brand personality, such as first-class, the best in the world, trustworthy, safe, and global. This was in accordance with a previous study that highlighted the relationship between brand personality and trust [60] and also provided empirical evidence for Celmer et al.'s framework of trust in branded driving automation [21].

**Characteristic 4: Have no negative cognition about the unethical event in drivers' cognitive structures.**   Focusing on Brand 3, the valid word number and different word numbers were both between the two brands in the high trust group, Brands 1 and 2, which implied that drivers' cognitive structures for Brand 3 were almost as rich and varied as Brand 1s and 2. Hence, Brand 3 could also be highly trusted. However, it was contradictory as Brand 3 was in

the medium trust group. Since Brand 3 was the only brand that has some associated words (23.6% of total associated words) relevant to unethical events, it was concluded that the negative cognition composed of these words could be invalid in drivers' cognitive structures to help build trust or could even reduce the trust built previously. According to previous studies on business behavior and consumer trust, unethical business behavior and climate of the employees and organizations were significantly related to lower levels of trust [29,30]. In contrast, ethical business behavior positively influenced consumers' trust [27,31]. Consequently, negative unethical events in the cognitive structure could lower drivers' trust in automobile brands and their products. Therefore, it might be necessary for brands to avoid the formation of drivers' negative cognition regarding such unethical events.

## Practice implications

Our findings have several practical implications for calibrating initial trust in Level 5 ADS. To avoid the disuse and misuse of Level 5 ADS, the influence of automobile brands should be considered during the initial trust calibration. For automobile corporations, determining consumers' cognition of their automobile brands and performing targeted brand management based on consumers' cognition is required. This may help eliminate consumers' distrust in their Level 5 ADS. In contrast, drivers' over-trust in Level 5 ADS in some highly trusted automobile brands may exist, which can lead to misuse, and thus, cause negative consequences, such as severe accidents. Therefore, for a more appropriate initial trust of drivers in Level 5 ADS, it is also necessary to pay attention to the prevention of drivers' over-trust in the automobile brands. Finally, considering the existing problems of distrust and over-trust in driving automation and common civil and market attributes of ADS with various SAE levels of driving automation [1], the practical implications may also be applicable to calibrating initial trust in ADS with other SAE levels of driving automation [1] besides Level 5 ADS.

## Limitations

First, this study selected only nine automobile brands to verify the hypotheses in Survey 1. In addition, only five brands were further selected in Survey 2. However, besides the selected brands, many other brands or companies have been developing driving automation. Therefore, the relationships between automobile brands, trust in automobile brands, initial trust in Level 5 ADS, and drivers' cognitive structures of automobile brands should be examined with more automobile brands. In contrast, since only one or two brands were selected for each trust group in survey 2, more conclusions regarding drivers' cognitive structures of automobile brands could not be drawn. Moreover, all the findings should be further confirmed with experimental research that can manipulate brand characteristics to observe the differences in the levels of initial trust in driving automation. Finally, this study focused only on the brands of traditional automobile manufacturers. Currently, the social realization of driving automation is also associated with entities other than traditional automobile manufacturers, such as technology and transportation service companies (e.g., Google and Uber), which also provide technological or service support for the development of driving automation. Therefore, research can focus on trust formation when a vehicle with ADS was manufactured and promoted by multiple entities, instead of only traditional automobile manufacturers.

## Conclusion

The level of drivers' initial trust in the Level 5 ADS depended on the automobile brand that developed the Level 5 ADS and could be positively influenced by the levels of drivers' trust in the automobile brands. For automobile brands that could earn higher levels of driver trust in

the automobile brand and Level 5 ADS, drivers possessed richer and more varied cognitive structures with some particular characteristics. Consequently, the influence of automobile brands and drivers' prior trust in automobile brands should be noted for the initial trust calibration of Level 5 ADS to avoid future disuse and misuse.

## Supporting information

**S1 File. Minimal underlying data set of Survey 1.**
(PDF)

## Acknowledgments

We thank our colleagues Suyang An and Fan Yang for their help with manuscript revision, and the anonymous reviewers for their comments that improved the manuscript.

## Author Contributions

**Conceptualization:** Zixin Cui, Makoto Itoh.

**Data curation:** Zixin Cui.

**Formal analysis:** Zixin Cui.

**Funding acquisition:** Makoto Itoh.

**Investigation:** Zixin Cui.

**Methodology:** Zixin Cui, Makoto Itoh.

**Project administration:** Zixin Cui.

**Resources:** Zixin Cui.

**Validation:** Zixin Cui, Nianzhi Tu, Makoto Itoh.

**Visualization:** Nianzhi Tu.

**Writing – original draft:** Zixin Cui, Nianzhi Tu.

**Writing – review & editing:** Zixin Cui, Nianzhi Tu, Makoto Itoh.

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
