## [Decision Letter · Decision Letter 0]

24 Jan 2023

PONE-D-22-29912Effects of brand and brand trust on initial trust in fully automated driving systemPLOS ONE

Dear Dr. Cui,

Thank you for submitting your manuscript to PLOS ONE. After careful consideration, we feel that it has merit but does not fully meet PLOS ONE’s publication criteria as it currently stands. Therefore, we invite you to submit a revised version of the manuscript that addresses the points raised during the review process. Please submit your revised manuscript by Mar 10 2023 11:59PM. If you will need more time than this to complete your revisions, please reply to this message or contact the journal office at plosone@plos.org. Please include the following items when submitting your revised manuscript:A rebuttal letter that responds to each point raised by the academic editor and reviewer(s). You should upload this letter as a separate file labeled 'Response to Reviewers'.A marked-up copy of your manuscript that highlights changes made to the original version. You should upload this as a separate file labeled 'Revised Manuscript with Track Changes'.An unmarked version of your revised paper without tracked changes. You should upload this as a separate file labeled 'Manuscript'.If applicable, we recommend that you deposit your laboratory protocols in protocols.io to enhance the reproducibility of your results. Protocols.io assigns your protocol its own identifier (DOI) so that it can be cited independently in the future. For instructions see: https://journals.plos.org/plosone/s/submission-guidelines#loc-laboratory-protocols. Additionally, PLOS ONE offers an option for publishing peer-reviewed Lab Protocol articles, which describe protocols hosted on protocols.io. Read more information on sharing protocols at https://plos.org/protocols?utm_medium=editorial-email&utm_source=authorletters&utm_campaign=protocols.

We look forward to receiving your revised manuscript.

Kind regards,

Wojciech Trzebinski, Ph.D.

Academic Editor

PLOS ONE

Journal Requirements:

2. Please provide additional details regarding participant consent. In the Methods section, please ensure that you have specified (1) whether consent was informed and (2) what type you obtained (for instance, written or verbal). If your study included minors, state whether you obtained consent from parents or guardians. If the need for consent was waived by the ethics committee, please include this information.

3. Please change "female” or "male" to "woman” or "man" as appropriate, when used as a noun (see for instance https://apastyle.apa.org/style-grammar-guidelines/bias-free-language/gender).

Additional Editor Comments:

Both reviewers raised several concerns related to your method. Additionally, Reviewer 2 expressed some comments about the theoretical part, including how you formulated your hypotheses. Note that the comments ot Reviewer 2 are included in the attached PDF. Please address the Reviewers' comments or at least respond in to them in the rebuttal letter.

Reviewers' comments:

Reviewer's Responses to Questions

**Comments to the Author**

1. Is the manuscript technically sound, and do the data support the conclusions?

Reviewer #1: Yes

Reviewer #2: Partly

2. Has the statistical analysis been performed appropriately and rigorously? 

Reviewer #1: Yes

Reviewer #2: Yes

3. Have the authors made all data underlying the findings in their manuscript fully available?

Reviewer #1: No

Reviewer #2: No

4. Is the manuscript presented in an intelligible fashion and written in standard English?

Reviewer #1: Yes

Reviewer #2: No

5. Review Comments to the Author

Reviewer #1: Thank you for the timely contribution. I only have a couple of questions/suggestions that the authors may want to consider.

1. Although the study doesn't focus primarily on dispositional trust, it would be interesting and helpful to see results cut by age and gender, as the samples include a wide age range and has a balanced gender split. Presenting such result may also help readers to better understand the possible linkage between dispositional and initial trust.

2. Driving automation is not only offered by automobile manufacturers, but also from technology companies focused on the software development, and also distributed through transportation service companies such as Uber and Lyft. I'm wondering how trust may form when there are multiple producing/delivery parties involved, and if there's anything that can be inferred on this topic from the study conducted.

Reviewer #2: Dear author,

Thank you for submitting your article for review. I have reviewed the document and have added my feedback to the attached PDF. If there are any you do not agree with, please provide a suitable rebuttal. Finally, the manuscript requires thorough proofreading to remove grammatical errors and words used in the wrong context.

Thank you for your time and I wish you the best of luck with your future research.

- - - - - - - - - - - - - - - - - - - - - - - - - - - - - - - - - - - - - - - - - - - - - - - - - - - - - - - - - - - - - - - - - - - - - - - - - - - - - - - - - - - - - - - - - - - - - - - - - - - - - - - - - - - - - - - - - - - - - - - - - - - - - - - - - - - - - - - - - - - - - - - - - - - - - - - - - - - - - - - - - - - - - - - - - - - - - - - - - - - - - - - - - - - - - - - - - - - - - - - - - - - - - - - - - - - - - - - - - - - - - - - - - - - - - - - - - - - - - - - - - - - - - - - - - - - - - - - - - - - - - - - - - - - - - - - - - - - - - - - - - - - - - - - - - - - - - - - - - - - - - - - - - - - - - - - - - - - - - - - - - - - - - - - - - - - - - - - - - - - - - - - - - - - - - - - - - - - - - - - - - - - - - - - - - - - - - - - - - - - - - - - - - - - - - - - - - - - - - - - - - - - - - - - - - - - - - - - - - - - - - - - - - - - - - - - - - - - - - - - - - - - - - - - - - - - - - - - - - - - - - - - - - - - - - - - - - - - - - - - - - - - - - - - - - - - - - - - - - - - - - - - - - - - - - - - - - - - - - - - - - - - - - - - - - - - - - - - - - - - - - - - - - - - - - - - - - - - - - - - - - - - - - - - - - - - - - - - - - - - - - - - - - - - - - - - - - - - - - - - - - - - - - - - - - - - - - - - - - - - - - - - - - - - - - - - - - - - - - - - - - - - - - - - - - - - - - - - - - - - - - - - - - - - - - - - - - - - - - - - - - - - - - - - - - - - - - - - - - - - - - - - - - - - - - - - - - - - - - - - - - - - - - - - - - - - - - - - - - - - - - - - - - - - - - - - - - - - - - - - - - - - - - - - - - - - - - - - - - - - - - - - - - - - - - - - - - - - - - - - - - - - - - - - - - - - - - - - - - - - - - - - - - - - - - - - - - - - - - - - - - - - - - - - - - - - - - - - - - - - - - - - - - - - - - - - - - - - - - - - - - - - - - - - - - - - - - - - - - - - - - - - - - - - - - - - - - - - - - - - - - - - - - - - - - - - - - - - - - - - - - - - - - - - - - - - - - - - - - - - - - - - - - - - - - - - - - - - - - - - - - - - - - - - - - - - - - - - - - - - - - - - - - - - - - - - - - - - - - - - - - - - - - - - - - - - - - - - - - - - - - - - - - - - - - - - - - - - - - - - - - - - - - - - - - - - - - - - - - - - - - - - - - - - - - - - - - - - - - - - - - - - - - - - - - - - - - - - - - - - - - - - - - - - - - - - - - - - - - - - - - - - - - - - - - - - - - - - - - - - - - - - - - - - - - - - - - - - - - - - - - - - - - - - - - - - - - - - - - - - - - - - - - - - - - - - - - - - - - - - - - - - - - - - - - - - - - - - - - - - - - - - - - - - - - - - - - - - - - - - - - - - - - - - - - - - - - - - - - - - - - - - - - - - - - - - - - - - - - - - - - - - - - - - - - - - - - - - - - - - - - - - - - - - - - - - - - - - - - - - - - - - - - - - - - - - - - - - - - - - - - - - - - - - - - - - - - - - - - - - - - - - - - - - - - - - - - - - - - - - - - - - - - - - - - - - - - - - - - - - - - - - - - - - - - - - - - - - - - - - - - - - - - - - - - - - - - - - - - - - - - - - - - - - - - - - - - - - - - - - - - - - - - - - - - - - - - - - - - - - - - - - - - - - - - - - - - - - - - - - - - - - - - - - - - - - - - - - - - - - - - - - - - - - - - - - - - - - - - - - - - - - - - - - - - - - - - - - - - - - - - - - - - - - - - - - - - - - - - - - - - - - - - - - - - - - - - - - - - - - - - - - - - - - - - - - - - - - - - - - - - - - - - - - - - - - - - - - - - - - - - - - - - - - - - - - - - - - - - - - - - - - - - - - - - - - - - - - - - - - - - - - - - - - - - - - - - - - - - - - - - - - - - - - - - - - - - - - - - - - - - - - - - - - - - - - - - - - - - - - - - - - - - - - - - - - - - - - - - - - - - - - - - - - - - - - - - - - - - - - - - - - - - - - - - - - - - - - - - - - - - - - - - - - - - - - - - - - - - - - - - - - - - - - - - - - - - - - - - - - - - - - - - - - - - - - - - - - - - - - - - - - - - - - - - - - - - - - - - - - - - - - - - - - - - - - - - - - - - - - - - - - - - - - - - - - - - - - - - - - - - - - - - - - - - - - - - - - - - - - - - - - - - - - - - - - - - - - - - - - - - - - - - - - - - - - - - - - - - - - - - - - - - - - - - - - - - - - - - - - - - - - - - - - - - - - - - - - - - - - - - - - - - - - - - - - - - - - - - - - - - - - - - - - - - - - - - - - - - - - - - - - - - - - - - - - - - - - - - - - - - - - - - - - - - - - - - - - - - - - - - - - - - - - - - - - - - - - - - - - - - - - - - - - - - - - - - - - - - - - - - - - - - - - - - - - - - - - - - - - - - - - - - - - - - - - - - - - - - - - - - - - - - - - - - - - - - - - - - - - - - - - - - - - - - - - - - - - - - - - - - - - - - - - - - - - - - - - - - - - - - - - - - - - - - - - - - - - - - - - - - - - - - - - - - - - - - - - - - - - - - - - - - - - - - - - - - - - - - - - - - - - - - - - - - - - - - - - - - - - - - - - - - - - - - - - - - - - - - - - - - - - - - - - - - - - - - - - - - - - - - - - - - - - - - - - - - - - - - - - - - - - - - - - - - - - - - - - - - - - - - - - - - - - - - - - - - - - - - - - - - - - - - - - - - - - - - - - - - - - - - - - - - - - - - - - - - - - - - - - - - - - - - - - - - - - - - - - - - - - - - - - - - - - - - - - - - - - - - - - - - - - - - - - - - - - - - - - - - - - - - - - - - - - - - - - - - - - - - - - - - - - - - - - - - - - - - - - - - - - - - - - - - - - - - - - - - - - - - - - - - - - - - - - - - - - - - - - - - - - - - - - - - - - - - - - - - - - - - - - - - - - - - - - - - - - - - - - - - - - - - - - - - - - - - - - - - - - - - - - - - - - - - - - - - - - - - - - - - - - - - - - - - - - - - - - - - - - - - - - - - - - - - - - - - - - - - - - - - - - - - - - - - - - - - - - - - - - - - - - - - - - - - - - - - - - - - - - - - - - - - - - - - - - - - - - - - - - - - - - - - - - - - - - - - - - - - - - - - - - - - - - - - - - - - - - - - - - - - - - - - - - - - - - - - - - - - - - - - - - - - - - - - - - - - - - - - - - - - - - - - - - - - - - - - - - - - - - - - - - - - - - - - - - - - - - - - - - - - - - - - - - - - - - - - - - - - - - - - - - - - - - - - - - - - - - - - - - - - - - - - - - - - - - - - - - - - - - - - - - - - - - - - - - - - - - - - - - - - - - - - - - - - - - - - - - - - - - - - - - - - - - - - - - - - - - - - - - - - - - - - - - - - - - - - - - - - - - - - - - - - - - - - - - - - - - - - - - - - - - - - - - - - - - - - - - - - - - - - - - - - - - - - - - - - - - - - - - - - - - - - - - - - - - - - - - - - - - - - - - - - - - - - - - - - - - - - - - - - - - - - - - - - - - - - - - - - - - - - - - - - - - - - - - - - - - - - - - - - - - - - - - - - - - - - - - - - - - - - - - - - - - - - - - - - - - - - - - - - - - -

6. PLOS authors have the option to publish the peer review history of their article (what does this mean?). If published, this will include your full peer review and any attached files.

Reviewer #1: No

Reviewer #2: No

---

## [Author Response · Author response to Decision Letter 0]

22 Mar 2023

Dear Reviewers,

Thank you very much for the comments and suggestions. We have further revised the manuscript for further improvement based on your comments. The major changes in the manuscript are as follows:

1.We have revised the sentences in “Introduction” to make clearer statements of the driving automation referred in the current study (Page 3, Line 30-35; Page 3-4, Line 47-52).

2.We have changed all of the “fully automated driving system (FADS)” to “Automated Driving System with full driving automation (Level 5 ADS)” based on the the taxonomy and definition of driving automation by the Society of Automotive Engineers (SAE 2021).

3.We have added data analyses against age and gender for study 1 (Page 12-13, Line 201-210 for the methods, Page 15, Line 245-249 and Page 15, Table 4 for the results).

4.We have added limitations of the current study (Page 29, Line 513-518) 

5.We have addressed the language problems throughout the manuscript to make clearer statements.

All of the changes made in the revised manuscript have been highlighted in yellow in the file "Revised Manuscript with Track Changes" except for the format and language issues. 

The followings are our reply to your comments one by one. 

Reviewer #1

Comment 1

Although the study doesn't focus primarily on dispositional trust, it would be interesting and helpful to see results cut by age and gender, as the samples include a wide age range and has a balanced gender split. Presenting such result may also help readers to better understand the possible linkage between dispositional and initial trust.

Reply 

Thank you for your suggestions.

For survey 1, we have added analyses against age and gender to test whether the effect of trust in automobile brand on trust in Level 5 ADS is invariant across gender and age. We updated the details of the methods (Page 12-13, Line 201-210). “Since people of different ages and gender can develop initial trust in driving automation based on different factors, such as cognitive or emotional factors towards driving automation [41], it was also necessary to verify whether the effect of trust in automobile brands on trust in Level 5 ADS was invariant across gender and age. For sex, a multigroup invariance test was performed, and the invariance was indicated by a significant difference in the chi-square value (p < .05) [40] or ΔCFI, ΔGFI, ΔAGFI, and ΔRMSEA, which were not higher than 0.01 [42]. For age, the invariance was indicated by the non-significant moderating effect of age on the relationship. Therefore, a “moderation by interaction terms” method [43] was used, in which age was first multiplied with the scores of each item of trust in the automobile brand, and the subsequent combined effect was examined against trust in Level ADS. ”

For gender, we added multigroup invariance test. The results showed the positive effect of trust in automobile brand on trust in Level 5 ADS was invariant for male (Std. estimate = 0.699, p < .001) and female (Std. estimate = 0.599, p < .001) (Table 4; Page 15, Line 245-247). 

Then for age, we added moderating effect test by “moderation by interaction terms” based on the previous structural equation modeling. The results showed the moderating effect of age was not significant on the relationship between trust in automobile brand and trust in Level 5 ADS (male: Std. estimate = -0.021, p = .682; female: Std. estimate = 0.025, p = .573) (Page 15, Line 248-249). 

However, because of the invariant results across age and gender for survey 1, we did not continue to add analyses against age and gender for survey 2.

41.Cui Z, Tu N, Lee J, Itoh M. Influence of Demographic Factors and Automobile Brands on the Structure of Initial Trust in Driving Automation. In: 2023 Transportation Research Board 102nd Annual Meeting. 2023.

42.Anderson JC, Gerbing DW. Structural equation modeling in practice: A review and recommended two-step approach. Psychological bulletin. 1988;103(3): 411-423. doi: https://doi.org/10.1037/0033-2909.103.3.411

43.Wu G, Hu Z, Zheng J. Role stress, job burnout, and job performance in construction project managers: the moderating role of career calling. International journal of environmental research and public health. 2019;16(13): 2394. doi: 10.3390/ijerph16132394 PMID: 31284496

Comment 2

Driving automation is not only offered by automobile manufacturers, but also from technology companies focused on the software development, and also distributed through transportation service companies such as Uber and Lyft. I'm wondering how trust may form when there are multiple producing/delivery parties involved, and if there's anything that can be inferred on this topic from the study conducted.

Reply 

Thank you for your comments.

We agree with your opinion about discussing trust formation when there are multiple producing/delivery parties involved. This is actually a limitation of our study. We have added sentences in the section “Limitation” (Page 29, Line 513-518) to make further statements of the limitations of the current study and possible research topic in the future. “Finally, this study focused only on the brands of traditional automobile manufacturers. Currently, the social realization of driving automation is also associated with entities other than traditional automobile manufacturers, such as technology and transportation service companies (e.g., Google and Uber), which also provide technological or service support for the development of driving automation. Therefore, research can focus on trust formation when a vehicle with ADS was manufactured and promoted by multiple entities, instead of only traditional automobile manufacturers. ”

Reviewer #2

Comment 1

The manuscript requires thorough proofreading to remove grammatical errors and words used in the wrong context. Page 2, line 17；Page 8 line 130; Page 9 line 157: “win” 

Reply

Thank you for your comments. 

We have changed "win" to "earn" in the three places and checked the same wrongs in other places such as Page 7 line 126; Page 8 lines 151; Page 24 line 404; Page 25 line 439; Page 26 line 443; Page 27 line 470; Page 29 line 522. Besides, we checked other language-usage issues.

Comment 2

Page 2 line 19 in abstract: Please revise this sentence “For automobile brands highly trusted in, drivers possess more rich and various cognitive structures with some particular characteristics. ”

Reply

Thank you for your suggestions. 

We have revised the sentence in abstract to “for automobile brands with higher trust in automobile brands and Level 5 ADS, drivers’ cognitive structures were richer and varied, which included particular characteristics.” (Page 2, Line 24-26)

Comment 3

Page 3 line 24: Should talk about SAE levels - you are referring to Level 5, which are a long way from being a reality

Reply

Thank you for your suggestions. 

We have revised the the first paragraph (Page 3, Line 30-35) to make clearer statements. Since the survey object is automated driving system with full driving automation (SAE level 5), we also changed all “FADS” to “Level 5 ADS” following the descriptions in the document “Taxonomy and Definitions for Terms Related to Driving Automation Systems for On-Road Motor Vehicles (J3016_202104)” from SAE International. “Following the taxonomy and definition of driving automation by the Society of Automotive Engineers [1], an Automated Driving System (ADS) with full driving automation (Level 5) can operate vehicles under all road conditions, just as a skilled human driver, and does not require any supervision. Although Level 5 ADS is a long way from reality, its practical application is always expected since it can further optimize the traffic environment and help decrease the risk rate of traffic accidents.”

Comment 4

Page 3 line 41: Based on your definition of FADs there are no conditions they operate as they are not a reality. I would limit the capability of what you define as a FAD to Level 4 in the introduction.

Reply

Thank you for your comments. 

We have rechecked the literature we referred and the descriptions of driving automation in the document “Taxonomy and Definitions for Terms Related to Driving Automation Systems for On-Road Motor Vehicles (J3016_202104)” from SAE International. We realized our mistakes in the understanding and presentation of FAD. Therefore, we deleted the inappropriate quotation of the previous studies, changed all “FADS” to “Level 5 ADS” and revised the sentences (Page 3-4, Line 47-52) to make clearer statements. “Level 5 ADS can work under all road conditions, just as a skilled human driver, and can reduce risk in unmanageable conditions, such as flooded roads and glare ice [1]. However, drivers’ over-trust and misuse of Level 5 ADS are problems that are impossible to predict. Therefore, the appropriate calibration of drivers’ trust in Level 5 ADS should be considered to ensure their appropriate use in the future.”

In addition, we added sentences in the section of “Practice implications” (Page 28 line 499-502) to further state that the practice implications from our findings may also be applicable for calibrating initial trust in Automated Driving System with other SAE levels of driving automation besides Level 5 ADS. “Finally, this study focused only on the brands of traditional automobile manufacturers. Currently, the social realization of driving automation is also associated with entities other than traditional automobile manufacturers, such as technology and transportation service companies (e.g., Google and Uber), which also provide technological or service support for the development of driving automation. Therefore, research can focus on trust formation when a vehicle with ADS was manufactured and promoted by multiple entities, instead of only traditional automobile manufacturers. ”

Comment 5

Page 9 line 160: You are assuming that all brands increases trust. Wouldn’t you assume that a strong respected brand would increase trust and the least respected brand would lower trust?

Reply

Thank you for your comments. 

We have revised the sentences (Page 8, Line 153-154) to make clearer hypothesis. “Hypothesis 1 (H1): There would be a positive relationship between drivers’ trust in automobile brands and their initial trust in Level 5 ADS. ”

Comment 6

Page 9 line 164: “ higher and less initial trust ” Doesn't make sense.

Reply

Thank you for your comments. 

We have revised the sentences (Page 9, Line 157-158) to make a clearer Hypothesis 3. “Hypothesis 3 (H3): The cognitive structures of automobile brand drivers with higher trust and lower trust would have distinctly different characteristics.”

Comment 7

Page 10 line 171: Was this survey in English or Japanese? Please state clearly.

Reply

Thank you for your suggestions.

We have added “in Japanese” to make a clearer state for both survey 1 (Page 9, line 162) and survey 2 (Page 17, line 285)

Comment 8

Page 10 line 179: “Brand 1 in the current paper. Brand 1, Brand 2”Please state these Brands e.g. Honda, Mercedes, etc.

Page 18 line 288: “ Brand 1 and 2 in the high trust group, Brand 3 and 5 ” Again, must make it very clear what these brands were

Page 24-25 line 383-392: Figure 5-9, Not readable

Reply

Thank you for your suggestions.

We did not revise these, because we have to comply with our ethics review results and hide the actual brands surveyed in the current study. For figure 5-9, the unreadable words with mosaics were those can easily lead readers to know what these brands are, such as the name of the product.

Comment 9

Page 10 line 187: “following”following what?

Reply

Thank you for your comments.

We have added the missed words “the descriptions of SAE” (Page 10 line 177).

Comment 10

Page 17 line 272: “The above results of repeated” Don't use the term above or below, state the section

Reply

Thank you for your comments.

We have revised the sentences to “The results of the repeated measures ANOVA on the main effect of automobile brand on trust showed both trust in automobile brand and trust in Level 5 ADS significantly differ across automobile brands.” to clearly state the section (Page 17 line 276-278）.

Comment 11

Page 17 line 283: “(58 males) ” Any females? 

Reply

Thank you for your comments.

We have checked the raw data and added the data of females (Page 17 line 287).

Comment 11

Page 17 line 283: “(M = 50.7 years,” Is this (Age) right, seems very high?

Reply

Thank you for your comments.

We have checked the raw data and revised the inaccurate or wrong values of age (Page 17 line 287). The accurate age range of the collected samples in study 2 was 25-70 years. Since there were 23.3% of 40-49 years, 33.0% of 50-59 years, and 22.3% of over 60 years, the averaged age was actually very high.

Comment 12

Page 33 line 548: “SAE. Taxonomy and definitions for terms related to on-road motor vehicle automated Driving Systems. 2018;J3016_201806. Available from: https://www.sae.org/standards/content/j3016_201806/”, Updated version: https://www.sae.org/standards/content/j3016_202104/

Reply

Thank you for your comments. 

We have updated the newest version in the reference list for introduction. “[1]SAE. Taxonomy and definitions for terms related to Driving Automation Systems for On-Road Motor Vehicle. 2021; J3016_202104. Available from: https://www.sae.org/standards/content/j3016_202104/. ”

For the one for method, because study 1 was conducted in 2020, during which the version of J3016_201806 was used to introduce driving automation to the participants, we added another reference in the reference list. “[37] SAE. Taxonomy and definitions for terms related to Driving Automation Systems for On-Road Motor Vehicle. 2018; J3016_201806. Available from: https://www.sae.org/standards/content/j3016_201806/.”

---

## [Decision Letter · Decision Letter 1]

5 Apr 2023

Effects of brand and brand trust on initial trust in fully automated driving system

PONE-D-22-29912R1

Dear Dr. Cui,

We’re pleased to inform you that your manuscript has been judged scientifically suitable for publication and will be formally accepted for publication once it meets all outstanding technical requirements.

Please see the comment of Reviewer #1 (below) asking you to explain how the L5 ADS was described to the respondents. I encourage you to insert this information into the final proof. Additionally, you may wish to clarify right before listing your hypotheses that they are based on your considerations presented above.

Kind regards,

Wojciech Trzebinski, Ph.D.

Academic Editor

PLOS ONE

Additional Editor Comments (optional):

Reviewers' comments:

Reviewer's Responses to Questions

**Comments to the Author**

1. If the authors have adequately addressed your comments raised in a previous round of review and you feel that this manuscript is now acceptable for publication, you may indicate that here to bypass the “Comments to the Author” section, enter your conflict of interest statement in the “Confidential to Editor” section, and submit your "Accept" recommendation.

Reviewer #1: All comments have been addressed

Reviewer #2: All comments have been addressed

2. Is the manuscript technically sound, and do the data support the conclusions?

Reviewer #1: Yes

Reviewer #2: Yes

3. Has the statistical analysis been performed appropriately and rigorously? 

Reviewer #1: Yes

Reviewer #2: Yes

4. Have the authors made all data underlying the findings in their manuscript fully available?

Reviewer #1: No

Reviewer #2: Yes

5. Is the manuscript presented in an intelligible fashion and written in standard English?

Reviewer #1: Yes

Reviewer #2: Yes

6. Review Comments to the Author

Reviewer #1: Thank you for addressing all previous comments. The only question I have left is how the L5 ADS was described to the respondents. Given that self-driving is a concept that is widely misunderstood by the general public, it would be helpful to see how the term was presented to participants and how a baseline understanding was established.

Reviewer #2: (No Response)

7. PLOS authors have the option to publish the peer review history of their article (what does this mean?). If published, this will include your full peer review and any attached files.

Reviewer #1: No

Reviewer #2: No

---

## [Editor Report · Acceptance letter]

26 Apr 2023

PONE-D-22-29912R1 

*Effects of brand and brand trust on initial trust in fully automated driving system*

Dear Dr. Cui:

I'm pleased to inform you that your manuscript has been deemed suitable for publication in PLOS ONE. Congratulations! Your manuscript is now with our production department. 

Kind regards, 

on behalf of

Dr. Wojciech Trzebinski 

Academic Editor

PLOS ONE